# New Molecular Mechanisms and Clinical Impact of circRNAs in Human Cancer

**DOI:** 10.3390/cancers13133154

**Published:** 2021-06-24

**Authors:** Giulia Fontemaggi, Chiara Turco, Gabriella Esposito, Silvia Di Agostino

**Affiliations:** 1Oncogenomic and Epigenetic Unit, IRCCS Regina Elena National Cancer Institute, Via Elio Chianesi 53, 00144 Rome, Italy; giulia.fontemaggi@ifo.gov.it (G.F.); chiara.turco@ifo.gov.it (C.T.); gabriella.esposito@ifo.gov.it (G.E.); 2Department of Health Sciences, “Magna Graecia” University of Catanzaro, 88100 Catanzaro, Italy

**Keywords:** circRNA, cancer, backsplicing, host gene, parental gene, translation of ncRNA, miRNA

## Abstract

**Simple Summary:**

Circular RNAs (circRNAs) belong to a new class of non-coding RNAs implicated in cellular physiological functions but also in the evolution of various human pathologies. Due to their circular shape, circRNAs are resistant to degradation by exonuclease activity, making them more stable than linear RNAs. Several findings reported that circRNAs are aberrantly modulated in human cancer tissues, thus affecting carcinogenesis and metastatization. We aim to report the most recent and relevant results about novel circRNA functions and molecular regulation, to dissert about their role as reliable cancer biomarkers, and to hypothesize their contribution to multiple hallmarks of cancer.

**Abstract:**

Next generation RNA sequencing techniques, implemented in the recent years, have allowed us to identify circular RNAs (circRNAs), covalently closed loop structures resulting in RNA molecules that are more stable than linear RNAs. This class of non-coding RNA is emerging to be involved in a variety of cell functions during development, differentiation, and in many diseases, including cancer. Among the described biological activities, circRNAs have been implicated in microRNA (miRNA) sequestration, modulation of protein–protein interactions and regulation of mRNA transcription. In human cancer, circRNAs were implicated in the control of oncogenic activities such as tumor cell proliferation, epithelial-mesenchymal transition, invasion, metastasis and chemoresistance. The most widely described mechanism of action of circRNAs is their ability to act as competing endogenous RNAs (ceRNAs) for miRNAs, lncRNAs and mRNAs, thus impacting along their axis, despite the fact that a variety of additional mechanisms of action are emerging, representing an open and expanding field of study. Furthermore, research is currently focusing on understanding the possible implications of circRNAs in diagnostics, prognosis prediction, effectiveness of therapies and, eventually, therapeutic intervention in human cancer. The purpose of this review is to discuss new knowledge on the mechanisms of circRNA action, beyond ceRNA, their impact on human cancer and to dissect their potential value as biomarkers and therapeutic targets.

## 1. Introduction

Many in vivo and in vitro studies and, more recently, analysis of liquid biopsy from cancer patients have shown that non-coding RNA (ncRNA), such as microRNA (miRNA) and long ncRNA (lncRNA), can be considered as good biomarkers for the diagnosis, prognosis and treatment of various human cancers [1,2]. Circular RNAs (circRNAs) were first detected by electron microscopy nearly 45 years ago and later confirmed to be present in the cytoplasm of eukaryotic cells [3,4]. Interestingly, through new RNA sequencing methodologies and new bioinformatics approaches, circRNAs have forcefully emerged in the clinical and basic research landscape [1,5,6]. These technologies highlighted that human transcriptome counts more than 180,000 circRNAs [7] and that their expression pattern varies among cell types, diseases and the developmental stages of living beings, including plants and invertebrates [8,9,10,11].

circRNAs are formed by a peculiar pre-mRNA that circularizes forming a covalently closed continuous loop through a process called “back-splicing”, in which a downstream 3′ splice donor site is joined with an upstream 5′ splice acceptor site. They can derive from spliced introns or from one or more exons, and sometimes they can have retained introns. Furthermore, they are poorly subjected to degradation by exonucleases as they lack 5′- and 3′- ends [12]. The splice event that generates a linear mRNA can compete with the back-splice mechanism [13]. circRNAs can be divided into three main categories: exonic circRNA (ecircRNA) consisting of one or more exons and representing 85% of all circRNAs, exon-intronic circRNA (EicirRNA) and circularized intronic RNA (ciRNA). The biogenesis mechanisms are briefly illustrated in the Figure 1. The characterization of circRNA groups is fully reviewed in [14].

Most circRNAs are exported to the cytoplasm, while intron-containing circRNAs are usually kept in the nucleus and affect the regulation of host-gene expression. Both EIciRNAs and ciRNAs can act as cis-regulatory molecules in the regulation of host-genes. EIciRNAs bind to the small nuclear ribonucleoprotein U1 (snRNP) to form the EIciRNAs-U1 snRNP complex, which combines with polymerase II (Pol II) to regulate the transcription promoter region of the host genes [15]. 

It has been reported that circRNA expression is globally downregulated in diverse human tumors such as colorectal and gastric adenocarcinoma, osteosarcoma, renal cell carcinoma, lung adenocarcinoma, hepatocellular carcinoma, and prostate cancer [16]. An explanation can be given by the fact that circRNAs are more stable than linear RNAs and that they can therefore accumulate in slow growing cells or in non-proliferating cells, while in the proliferating cells they are distributed and diluted in the progeny cells [7,17]. However, over-expressed circRNAs are also observed in human tumors and support the maintenance of the tumor phenotype. Interestingly, the distinct evaluation of expression and function between circRNAs and their linear RNAs showed that circRNAs are not mere by-products of splicing, but they play an important role in carcinogenesis independently of their linear transcripts [7,9,16].

Diverse biological functions of circRNAs have been reported. In the scientific literature, the most reported function is that in which the circRNAs act as “sponges” for miRNAs, lncRNAs and some mRNAs, affecting the functions of their target genes [5,9]. This review aims to discuss other circRNA functions that are emerging. circRNAs can bind to specific RNA binding proteins (RBPs), sequester specific protein factors, and encode proteins/peptides that are involved in carcinogenesis and metastatization [16,17]. Recently, it has been shown that circRNAs are localized in exosomes, and therefore they are very stable in biological fluids. Table 1 summarizes some of the recent oncogenic and tumor suppressor circRNAs discussed in this review. 

New studies have revealed the roles of exosomal circRNAs in the onset of cancer [64]. From this point of view, circRNAs are considered novel and promising biomarkers for the diagnosis and progression of cancer, for the evaluation of chemotherapy treatments and perhaps as new tools for targeted molecular therapies.

## 2. A Variety of circRNA/RBP Complexes with Diverse Functions: Beyond miRNA Sponging Activity

The molecular properties of circRNAs extend beyond their miRNA sponging ability. Many studies have reported that groups of circRNAs serve as protein decoys, scaffolds and recruiters in various physiological and pathological contexts. Similar to linear lncRNAs, circRNAs may interact with single or multiple proteins to influence their cellular functions, thereby regulating gene transcription, inhibiting cell cycle, and promoting proliferation and cell survival, which may explain, moreover, their involvement in tumor progression [65]. Some known functions are schematically illustrated in the Figure 2 and described in the following paragraphs.

### 2.1. circRNAs’ Ability to Interact with Proteins and to Function as Protein Decoys

Recent studies indicate that circRNAs can serve as decoys that compete with mRNAs in binding to proteins. RNA–protein interactions can influence proteins’ expression and function. Usually, RNAs, similarly to DNA, interact with proteins through base stacking, hydrogen bonding, and hydrophobic interactions with van der Waals contacts. However, RNA–protein interactions significantly depend on the tertiary structure of the RNA molecules. In this regard, circRNAs are able to form different tertiary structures on the bound protein. By binding to proteins, circRNAs may favor or inhibit their activities, modulating various aspects of cell physiology. Emerging studies have demonstrated that, through direct interaction, circRNAs may also act as protein recruiters in a precise subcellular location, also impacting their abundance in a specific cellular compartment [66,67,68].

An example of positive regulation of function is circAmotl1, a circular RNA originating from exon 3 of the angiomotin-like 1 (AMOTL1) gene. circAmotl1 interacts with c-Myc protein, facilitating its nuclear translocation and enhancing cancer growth. circAmotl1 is also able to recruit Signal Transducer and Activator of Transcription 3 (STAT3) in the nucleus to stabilize STAT3 binding to the DNA-methyltransferase promoter [18,19].

On the contrary, circRNAs have been also shown to be able to bind and sequester various protein factors, preventing their functions. For example, circFoxo3 has been shown to bind to a variety of proteins in different cells. It can interact with anti-stress proteins ID-1, FAK, HIF-1a, and the transcription factor E2F1, thereby retaining these proteins in the cytoplasm and leading to cellular senescence [20].

Another example of negative regulation by circRNA is that of circPABPN1. In human cervical carcinoma, high levels of circPABPN1 suppressed HuR binding to *PABPN1* mRNA, inhibiting *PABPN1* translation and resulting in decreased cell proliferation [21].

Very recently, interaction of circPCNX (hsa_circ_0032434) with AUF1 protein has been shown to selectively prevent AUF1 binding to p21 mRNA, leading to enhanced p21 mRNA stability and p21 protein production, thereby suppressing cell growth in human cervical carcinoma cells [22].

These recent papers, well described as circRNA–protein interactions, can influence proteins’ expression and function.

### 2.2. circRNAs Enhancing Protein Complexes Formation and/or Activity

Many studies have shown that circRNAs can function like lncRNAs in facilitating the formation of protein complexes. circRNAs are able to interact with proteins and mRNAs and form ternary complexes, which can regulate mRNA stability or control translation. The functional output of these complexes in cancer cells is frequently the modulation of cell plasticity, namely the stem potential, and the transition between the epithelial and mesenchymal states. An example of circRNA impinging on stem potential is circMALAT1, which is produced by back-splicing of lncRNA MALAT1. circMALAT1 is highly expressed in cancer stem cells (CSCs), where it constitutes a ternary complex with the mRNA of paired box 5 (PAX5), a tumor suppressor, and ribosomes. This causes the block of PAX5 mRNA translation and the maintenance of the stem state in hepatocellular carcinoma [23].

Another circRNA, which contributes to cell plasticity in cancer cells is circNSUN2. Specifically, circNSUN2 has a role in the induction of epithelial-mesenchymal transition (EMT) and the promotion of colorectal cancer (CRC) aggressiveness. This circRNA combines with IGF2BP2 protein and high mobility group A2 (HMGA2) mRNA, forming a ternary complex able to stabilize the mRNA [24]. 

YAP is the key component of Hippo pathway, which plays a crucial role in tumorigenesis. Interestingly, YAP expression can be regulated by its circRNA, circYAP, through the inhibition of YAP translation initiation machinery. Particularly, circYAP can bind to its linear counterpart YAP mRNA and to the translation initiation associated proteins, eIF4G and PABP. This complex abrogates the interaction of PABP with the poly(A) tail and of eIF4G with the 5′-cap of the YAP mRNA, finally preventing YAP translation initiation. [25].

An example of circRNA that favors the functional activation of a complex between two proteins rather than the formation of this complex has been recently reported. It is the case of circPOK, an oncogenic circRNA mainly expressed in mesenchymal cells. circPOK is in antithesis with its linear counterpart, which suppresses tumor progression. circPOK interacts with interleukin enhancer binding factor 2/3 (ILF2/3) protein complex in the nucleus and affects the functionality of the complex. Specifically, circPOK-bound ILF2/3 complex binds with increased affinity to the mRNAs encoding for proangiogenic factors such as IL-6 and VEGF, providing a proliferative advantage and augmented angiogenic potential [26]. These studies confirmed that, in ternary complexes, circRNA–protein interactions are quite common; nevertheless, their exact structures and precise processes of assembly and disassembly have not been completely clarified. 

### 2.3. circRNA Permitting the Interaction between Different Proteins

Recently, it was discovered that circRNAs have a role in the interaction between different proteins. Three modes of circRNA–protein interaction are currently known: circRNA binds to both proteins to reinforce their interaction; circRNA binds to protein A to modulate, positively or negatively, its interaction with protein B; circRNA dissociates proteins that had originally combined with each other [69].

circFoxo3 is one of the most well-studied circRNA in the context of cancer [2]. It originates from *FOXO3* gene and can regulate stress resistance, cell apoptosis and the cell cycle. Foxo3 protein is downregulated in many tumors and is considered a tumor suppressor gene [70]. The circular form of Foxo3, circFoxo3, has demonstrated tumor suppressor as well as oncogenic potential, depending on the considered cancer context. In breast cancer, circFoxo3 acts as a tumor suppressor by increasing the Foxo3 protein level while repressing p53 expression. By binding both p53 and MDM2 (the E3 ubiquitin-protein ligase that causes p53 degradation), circFoxo3 promotes MDM2-induced p53 ubiquitination and subsequent degradation. At the same time, circFoxo3 prevents MDM2 from inducing Foxo3 ubiquitination and degradation, with consequent increased levels of the Foxo3 protein. As a result, cell apoptosis is induced by upregulation of the Foxo3 downstream target PUMA [68].

circFoxo3 may also enhance the interaction between p21 and Cdk2 proteins in the cytoplasm, thereby preventing Cdk2 interaction with Cyclins A and E and subsequently blocking cell cycle progression. Therefore, circ-Foxo3 can control cell-cycle entry, forming a complex with p21 and Cdk2 [27].

A single circRNA can influence the interactions between proteins both positively and negatively, as exemplified by circCcnb1. In a wild-type *TP53* context, circCcnb1 can bind H2AX and wt-p53 proteins, enabling wt-p53-cell proliferation and survival. On the contrary, in mutant p53-carrying cells, circCcnb1 forms a complex with H2AX and Bclaf1 proteins, facilitating cancer cell death. This occurs because wt-p53 has a greater affinity for H2AX compared with mutant p53 and then leaves Bclaf1 free to bind to Bcl2, facilitating cell proliferation. Mutant p53, however, is unable to bind to H2AX, so Bclaf1 can interact with H2AX and circCcnb1 to induce the death of mutant p53-carrying cancer cells [28].

Finally, circRNA can sequester proteins, blocking their interaction with other proteins. A recent study reports that circANRIL, which is transcribed at a locus of atherosclerotic cardiovascular disease on chromosome 9p21, confers atheroprotection by controlling ribosomal RNA maturation. circANRIL can bind Pescadillo ribosomal biogenesis factor 1 (PES1), disturbing its assembly with two other proteins, BOP1 and WDR12, which forms a PeBoW complex and is important for 60S-preribosomal assembly. The inhibition of rRNA maturation leads to compromised ribosome biogenesis and induces nucleolar stress as well as p53 stabilization. Therefore, this leads to an increase in apoptosis and a decrease in proliferation that will detain the development of atherosclerosis [29]. Another study reports that circGSK3β can promote ESCC migration and invasion by decreasing β-catenin phosphorylation by GSK3β and its sequential ubiquitination [30]. 

These recent evidence shows that circRNAs play a role in the establishment of functional platforms where protein molecules can interact. 

### 2.4. Transcriptional Regulation

circRNAs exert many functional roles in development and disease, and among these is the regulation of gene expression. In the tumor context, some circRNAs have been shown to regulate gene transcription. This mainly occurs through the direct recruitment of the circRNA on the promoter region of a target gene; this recruitment impinges of the accessibility of the promoter and on the transcriptional activation rate.

We show here two examples referring, respectively, to gastric (GC) and liver (HCC) cancer. circDONSON is a circRNA able to directly control the expression of SOX4, a transcription factor often elevated in human cancers, where it generally correlates with tumor-progression and poor disease outcome. It has been observed that circDONSON is enriched on the SOX4 promoter region (~600/400 bp upstream of the transcription start site). The silencing of circDONSON suppresses the active transcription mark H3K27ac on the SOX4 promoter, leading to a significant reduction of SOX4 expression. Importantly, the restoration of SOX4 leads to a reversal of the effects of circDONSON silencing on apoptosis, migration and invasion in gastric cancer cells. Data reported by Ding et al. (2019) demonstrate that circDONSON activates SOX4 transcription to promote GC progression [31].

Another example of a circRNA that impinges on gene transcription is circRHOT1, which regulates the expression of NR2F6 in HCC cells [32]. NR2F6 is highly expressed in HCC cells, and its expression requires circRHOT1, which is enriched in the NR2F6 promoter region and positively regulates, similarly to what has been observed in GC for circDONSON, its H3K27ac modification. The presence of circRHOT is necessary for the binding of RNA pol II and for the accessibility of the NR2F6 promoter. Specifically, circRHOT1 recruits the TIP60 protein to initiate NR2F6 expression. Accordingly, a positive correlation is present between the expression of NR2F6 and circRHOT1, as well as between NR2F6 and TIP60, in HCC tissues. All these elements highlight that circRHOT1 is essential for TIP60-mediated NR2F6 expression in HCC tissues. Functionally, circRHOT1 promotes the cell proliferation, migration and progression of HCC through NR2F6 activation [32]. 

These and many other papers strongly suggest that circRNAs may play a transcriptional regulatory role, the detailed mechanisms of which are discussed in the following sub-paragraphs. These studies provide important information for future exploration of the function of circRNAs transcriptional control in cancer.

### 2.5. Control of the Expression of Parental Genes: Control of Transcription Initiation

There is a class of circRNAs that localize in the nucleus, are associated with RNA polymerase II, and are able to regulate the expression of their parental genes in human cells. These circRNAs are called exon-intron circRNAs or EIciRNAs because exons are circularized with introns “retained” between exons (see also Figure 1).

Some circRNAs have been identified as containing exons and introns capable of regulating gene transcription in cis by forming specific interactions with snRNP U1 (small nuclear ribonucleoprotein) thanks to the binding site of snRNA U1 present in the introns [15]. snRNP U1 is a component of the splice complex that specifically binds to TFIIH, which is a general transcription initiator [33]. In particular, the authors identified two exon-intron circular RNAs, circ-EIF3J and circPAIP2, which can substitute snRNP U1 functions to create an additional complex with RNA Pol II and ensure the expression of their parental genes. Notably, depletion of these circRNAs decreased the transcription level of the corresponding EIF3J or PAIP2 host genes. RNA-DNA double fluorescence in situ hybridization (FISH) revealed that EIciEIF3J and EIciPAIP2 colocalized with the EIF3J and PAIP2 gene promoters, respectively. Collectively, these results suggest that some circRNAs could regulate the transcription of their parental genes in the transcription initiation phase. circRNA production could compete with pre-mRNA maturation; however, once produced, EIciRNAs can establish positive feedback loops for their own expression and that of their parental genes. However, circEIF3J and circPAIP2 were also linked to other gene loci, hypothesizing that these circRNAs may exert trans effects on other loci as well [15]. 

circRNAs can also affect the expression of parental genes by recruiting proteins to specific regions of the target promoters or by sponging proteins to prevent their binding. The proteins in question are: RNA-binding proteins (RBP), transcription factors, DNA demethylases and DNA methyltransferases. Promoter regions are the most studied regions for transcription regulation. There are many examples of circular RNAs that control the expression of their own gene in this way and thus favor tumor progression.

Feng et al., for example, identified a circRNA involved in the progression of prostate cancer (PC), circXIAP (hsa_circ_0005276), derived from three exons of XIAP gene. circXIAP and its host XIAP gene are both upregulated in PC. circXIAP, which has both cytoplasmic and nuclear localization, is able to recruit the RNA binding protein FUS to the XIAP gene promoter region in the nucleus and activate XIAP transcription, thereby promoting the proliferation, migration and epithelial-mesenchymal transition (EMT) of PC cells [34]. Another such example is FECR1, a circRNA consisting of FLI1 4-2-3 exons, which is able to recruit TET1 DNA demethylase on the FLI1 promoter inducing DNA demethylation, thus promoting greater expression of FLI1 gene and consequently favoring the invasion of breast cancer cells [35]. circCUX1 (hsa_circ_0132813), highly expressed in neuroblastoma, is located in the nucleus, consists of exon 2 and partial intron 2 of CUX1 gene, and interacts with EWS RNA-binding protein 1 (EWSR1), inhibiting the EWSR1-mediated transactivation of MYC-associated zinc finger protein (MAZ) and promoting CUX1 transcription, glycolysis and neuroblastoma progression [36].

### 2.6. Control of the Expression of Parental Genes: Control of Transcription Elongation

Transcription can also be regulated at the level of elongation. Circular RNAs, in particular intronic circRNAs (ciRNAs), are able to regulate transcription in the elongation phase. One of these identified ciRNAs was ci-ankrd52 [37]. Ci-ankrd52 originates from the second intron of the ankrd52 gene, encoding a protein with an unknown function. Synthetic antisense oligodeoxynucleotides (ASOs) targeting intron-derived ci-ankrd52 were able to downregulate the expression of these circular RNAs, also leading to a decrease in parental mRNA levels. The specific ASO for ci-ankrd52, being complementary to the pre-mRNA intron ankrd52, can bind to pre-mRNA and determine its degradation with the consequent decrease of mRNA levels. The co-expression of ASO and corresponding intronic RNAs, except ci-ankrd52, was unable to reduce ankrd52 mRNA levels, confirming the specificity of ci-ankrd52 in the regulation of its parental mRNA expression. A similar mechanism has also been reported for ci-mcm5 and ci-sirt7. Throughout in vitro pull-down assay, it was found that biotinylated ci-ankrd52 interacted with phosphorylated Pol II RNA. By using similar assays, it was possible to deduce that ci-mcm5 and ci-sirt7 also interact with Pol II. Pol II phosphorylation is a key step in the transcription elongation process [71]. The association of ciRNA with phosphorylated Pol II provided confirmation of its role as a regulator in transcription elongation.

### 2.7. Regulation of Protein Scaffold in Impacting Translation

RBPs are proteins able to interact with specific sequences of RNA molecules, regulating their maturation, transport, localization and translation. These proteins also participate in the formation of ribonucleoprotein complexes [72]. Bioinformatic analyses of circRNA sequences have documented low enrichment for RBP binding sites; however, experimental data suggest that circRNAs are able to bind to RBPs through specific binding sites. Since RNA-RBP interactions are significantly influenced by the tertiary structure of RNA molecules, the tertiary structure of circRNAs can exert an effect on their ability to bind proteins. 

EcRNAs may play a role in the regulation of translation by interacting with RBPs in the cytoplasm. Hsa_circ_0075804, for example, promotes the stability of transcription factor E2F3 mRNA by binding to HNRNPK, thereby promoting retinoblastoma proliferation in an E2F3-dependent manner [38]. Another circRNA, circBACH1, binds to HuR protein, thereby inducing its translocation from the nucleus to the cytoplasm, where HuR interrupts the Internal Ribosome Entry Sites (IRES) function in the 5′-UTR of p27 mRNA. This event results in inhibition of p27 translation and consequently in the promotion of HCC growth [39].

## 3. Translation of circRNAs in Human Cancer

Untranslated regions (UTR) 5′ and 3′ are considered essential elements that control translation in eukaryotic cells. Usually, initiation phase requires a 7-methylguanosine cap (m7G) at the 5′ end of mRNA. As circRNAs lack 5′ and 3′ ends, they have long been considered to have no coding capacity. Mounting evidence from recent literature, however, suggests unexpected protein-coding potential for circRNAs. Diverse circRNAs have indeed been shown to be associated with polysomes and to contain the AUG initiation codon as well as an open reading frame (ORF) of sufficient length. Specifically, it has been shown that translation of circRNAs occurs through cap-independent mechanisms [73]. Cap-independent translation was previously reported to control the expression of several mRNAs, especially under cellular stress conditions. Cap-independent translation initiation on circRNAs might be mediated by: (1) the presence of specific sequences, referred to as Internal Ribosome Entry Sites (IRES), which can directly recruit ribosomes to initiate translation; and (2) the presence of methylated adenosine nucleotides in the form of N6-methyladenosine (m6A), which can directly bind eIF3. Notably, it is highly reasonable that these two cap-independent translation pathways work coordinately. Indeed, a high m6A methylation level was detected in IRES-containing circZNF609, which is translated and functions in myogenesis [40]. m6A-driven translation requires initiation factor eIF4G2 and m6A reader YTHDF3; moreover, it is enhanced by methyltransferases METTL3/14, inhibited by demethylase FTO, and upregulated upon heat shock (Figure 3). Interestingly, in a high throughput screening, m6A-driven translation of circRNAs has been found to be widespread, with hundreds of endogenous circRNAs possessing translation potential [74]. Bioinformatics tools for identifying circRNA coding potential have been developed [73], and databases of translatable circRNAs have been built [75]; this will also contribute to giving a strong boost to the already active research on translatable circRNAs function.

A variety of circRNAs with coding potential have been reported as altered in human cancer. In Hepatocellular Carcinoma (HCC), a circRNA arising from β-catenin (circβ-catenin), which allows for the translation of a 370 amino acid peptide (β-catenin-370aa), has been identified [41]. Interestingly, β-catenin-370aa inhibits GSK3β by acting as a decoy and blocks GSK3β-induced β-catenin degradation, finally favoring β-catenin activity. Another interesting example of translated circRNA is circFBXW7, which encodes a 185 amino acid peptide (FBXW7-185aa). circFBXW7 has been reported in a variety of cancer types to act as a tumor suppressor, also through its ability to sponge microRNAs. The translated FBXW7-185aa has been so far identified in glioma and in triple-negative breast cancer [42,43], where it contributes to inhibiting proliferation and migration. Mechanistically, FBXW7-185aa induces the expression of FBXW7 and causes c-Myc degradation thanks to its ability to bind and inhibit USP28, a deubiquinating enzyme responsible for c-Myc stabilization.

## 4. Regulation of Immune Functions by circRNA

circRNAs have been reported to be able to control immunity in various ways. First of all, differently from linear RNA, which leads to robust cellular immune response, foreign circRNAs have reduced immunogenicity [76]. Importantly, the innate immune response is inhibited by N6-methyladenosine (m6A) RNA modification on human circRNAs. Indeed, m6A reader YTHDF2 sequesters m6A-circRNA and is essential for suppression of innate immunity [77].

In the context of cancer, circRNAs may have an impact on immune escape via different mechanisms. The most direct mechanism is their ability to impinge, mainly through sponging activity, on miRNAs that control the translation of immune checkpoint proteins such as PD1-PDL1 and CTLA4-CD80/86 (Figure 3). One such example is hsa_circ_0020397, a circRNA up-regulated in colorectal cancer (CRC) that controls cancer cell behavior by inhibiting miR-138 with subsequent enhancement of PD-L1 expression [44]. circRNAs have been also shown to control the expression of inflammatory cytokines in cancer, thus modulating the recruitment of various cell types and the composition of the tumor microenvironment. High level of circ_0020710, for example, upregulates CXCL12 expression by sponging miR-370-3p in melanoma cells. Accordingly, elevated circ_0020710 expression correlates with cytotoxic lymphocyte exhaustion, and, of note, a combination of a CXCL12/CXCR4 axis inhibitor and anti-PD-1 significantly attenuates tumor growth [45]. Another interesting example is circMET (hsa_circ_0082002), a circRNA enclosed in chromosome region 7q21-7q31, the amplification of which is associated with tumor recurrence and multidrug resistance in hepatocellular carcinoma (HCC). circMET expression promotes HCC development by inducing an epithelial to mesenchymal transition and enhancing an immunosuppressive tumor microenvironment. Specifically, circMET enhances the expression of Snail by sponging miR-30-5p; Snail in turn activates DPP4, which downregulates CXCL10, negatively regulating lymphocyte trafficking [46].

## 5. Functions of circRNAs in the Exosomes

Exosomes are small vesicles of 30–100 nm in diameter, surrounded by a phospholipid membrane, and originating through a process of endocytosis of the early endosome. They are secreted by most cells in both physiological and pathological conditions. They usually contain peptides, small proteins, mRNAs and non-coding RNAs that can regulate the behavior of recipient cells. They are present in fluids throughout the body and can be used as biomarkers for diagnosing human pathologies [78,79]. The tumor uses exosomes as a means of communication with the surrounding cells forming the tumor microenvironment to drive cancer proliferation, angiogenesis, invasion, metastasis and immune escape [80,81,82].

Recently, circRNAs have also been identified to be contained in the exosomes, and thanks to their resistance to degradation, they can be stably present during the travel of exosomes in bodily fluids that are transported over great distances. Several studies are underway to investigate the functions of exosomal circRNAs in tumorigenesis. The first study on exosomal circRNAs, which is very recent, analyzed circRNAs by using RNA-seq analyses of ribosomal RNA-depleted total RNA from MHCC-LM3 liver cancer cells and cell-derived exosomes [83]. They found that the ratio of circular (circRNA) to linear (mRNA) expression in exosomes was about six times higher than in cells. This suggested that circRNAs are more efficiently incorporated into exosomes than linear RNAs. The authors extended this analysis by validating the pool of circRNAs found in liver cancer by performing RT-qPCR analyses from exosomes purified from colon, lung, stomach, breast, and cervical cancer cells. Diverse studies have suggested that circRNAs are actively incorporated into exosomes with mechanisms that have not yet been elucidated, probably implicating pathways common to linear RNAs, as through specific sequences recognized by RNA binding proteins [83,84,85].

It has been hypothesized that the presence of circRNAs in the exosomes can be directly regulated by miRNA expression in donor cells. For example, CDR1-AS circRNA binds to miR-7 and strongly represses its activity towards target mRNAs [47,48]. CDR1-AS expression is significantly downregulated in the exosomes, but it rises under overexpressed miR-7 conditions [83]. This could suggest that, at least in part, the recruitment of circular RNAs into exosomes could be regulated by the expression level of the miRNAs that bind to them. Moreover, exosomal CDR1-AS was the first circRNA for which a function in tumor proliferation was identified. It can induce growth suppression in the recipient cell by sequestering endogenous miR-7 through sponge activity [83]. 

After the discovery by Li and colleagues, many papers have identified the presence of circRNAs in exosomes from cell lines and biological fluids in many diseases, including cancer. In tumors, they have been related not only to proliferation but also to several additional oncogenic processes, such as epithelial-mesenchimal transition (EMT), invasion, metastases and resistance to the therapies. 

Regarding the invasion process, circRNAs in exosomes, in very recent papers, are reported to render cancer cells more invasive, also by altering the tumor microenvironment. circ-PDE8A expression was detected by microarray analysis from liver-metastatic pancreatic ductal adenocarcinoma (PDAC) tissues, and high circ-PDE8A expression was correlated with lymphatic invasion, TNM stage and a poor survival rate of PDAC patients. The authors found the circPDE8A expression in plasma exosomes of PDAC patients and exosomal circPDE8A was associated with progression and prognosis in PDAC patients. Interestingly, circPDE8A promoted the EMT in recipient cells by activating the MACC/MET/ERK or AKT pathway [49].

Further evidence about the role of circRNAs in metastases of PDAC patients has been reported by Li and colleagues and very recently reviewed by Limb and colleagues using a meta-analysis approach [86]. circIARS expression was found to be up-regulated in pancreatic cancer tissues and in the plasma exosomes of patients with metastatic disease. In vitro experiments demonstrated that circ-IARS, through exosomes, promoted tumor invasion and the metastasis of HUVEC cells. High levels of circ-IARS are correlated with liver metastasis, vascular invasion, and the TNM stage and negatively correlated with survival time after surgery. circIARS is able to up-regulate RhoA and RhoA-GTP levels, increasing F-actin and focal adhesion and inducing endothelial monolayer permeability [50].

It is emerging that exosomal circRNA can induce cancer cell invasion not only by making cancer cells more invasive but also by altering the tumor microenvironment.

In prostate cancer, a very recent study showed the gene expression analysis and the screening of circRNAs from a cohort of 144 PCa patients who went into recurrence after surgery [51]. Using the results, the authors developed a new circRNA risk score model to predict prognosis. The Gene Ontology analysis of differentially expressed mRNAs between high and low risk score PC patient groups identified a group of eight circRNAs (circ_178252, circ_115617, circ_14736, circ_30029, circ_117300, circ_176436, circ_112897, and circ_17720) related to recurrence involving pathways, which belong to the tumor microenvironment (TME). Often, the PC recurrent cases are inevitably destined to evolve into metastases, marking a worse prognosis [87]. Due to the complex signaling and molecular mechanisms that are responsible for PC development and progression, the existing methods of prognosis prediction are still insufficient. This study could be an excellent starting point to implement some prediction molecular models for early detection, individualized therapeutic strategy, and distinguishing high-risk patients, not only in pancreatic cancer but also in other tumors where circRNAs have been found to be circulating.

Very recently, it was reported that circRNAs can participate in the occurrence and progression of lung cancer by exosome-transport. circSATB2 is highly expressed in non-small cell lung carcinoma (NSCLC) cells and tissues [52]. Regarding the molecular mechanism, circSATB2 acts by negatively regulating miR-326 expression and thereby inducing the expression of the fascin homolog 1, actin-binding protein 1 (FSCN1) [52]. Furthermore, circSATB2 is transferred by exosomes, promoting the proliferation, migration and invasion of NSCLC cells and also inducing the abnormal proliferation of normal human bronchial epithelial cells. Importantly, circSATB2 was found to be highly present in the serum exosomes of lung cancer patients with high specificity of correlation to lung cancer metastases, making it an interesting potential biomarker for the diagnosis of NSCLC [52]. 

Lung adenocarcinoma (LUAD) and lung squamous cell carcinoma are the main histologic subtypes of NSCLC. The incidence of LUAD has been observed to be increasing, year after year, eventually becoming the most common subtype of NSCLC [88]. Lin and colleagues examined, using high-throughput whole transcriptome sequencing, circRNA expression in serum exosomes isolated from advanced-stage LUAD patients [89]. In silico analysis of the putative pathways affected by the differentially expressed circRNAs showed that tumor-related signaling pathways were the most represented. The strongest correlation was reported to be with vascular endothelial growth factor (VEGF) signaling, the major regulator of angiogenesis, and with promising signaling pathways to be targeted with therapies already in use in clinical practice [89].

Ovarian cancer (OC) ranks as the fifth leading cause of malignancy-associated mortality in women [90]. The ovary lies deep in the abdomen, and this anatomical position favors late diagnosis of cancer, metastases mainly in the peritoneal cavity and low survival rates of patients [91]. Therefore, research into biomarkers for early detection and prediction of metastasis has received extensive attention. Recently, circWHSC1 (circ_0001387) was detected in OC tissues at higher levels than in normal tissues without being already reported in other tumors [53]. Furthermore, Zong and colleagues reported that circWHSC1 expression was higher in moderately and poorly differentiated ovarian cancer tissues than in well-differentiated ones, suggesting a role for circWHSC1 as a histological biomarker in OC. Overexpression of circWHSC1 increased cell proliferation, migration and invasion by in vitro assays by sponging miR-145, a well-known tumor suppressor in various malignancies [53,92]. circWHSC1 was also found in exosomes secreted by OC cells; therefore, the authors explored the role of exosomal circWHSC1 in tumor peritoneal dissemination. Interestingly, mice injected with circWHSC1-exosomes exhibited a higher number of tumor nodules in the abdominal cavity than control mice, a loss of the original tight structure between cells in the peritoneal mesothelium and expression of EMT markers [53]. Very recently, high circWHSC1 expression was also observed in hepatocellular carcinoma tissues, where high circWHSC1 is associated with worse overall survival, indicating that circWHSC1 is also a powerful diagnostic indicator for HCC [93].

The results of these studies are in agreement with recent emerging evidence showing that circRNAs could be related to the tumor microenvironment, involved in tumor surveillance, angiogenesis, hypoxia, remodeling of the extracellular matrix (ECM), and other functions [94,95,96]. 

The circRNA functions in the exosomes reviewed in this paragraph are summarized in Table 1.

## 6. circRNAs in the Drug Resistance 

circRNAs, in addition to being hypothesized as biomarkers for early diagnosis, metastasis and prognosis, could also be valid tools in the detection of drug resistance. The potential mechanisms of circRNAs in tumors that exhibit resistance to drugs and radiotherapies are multiple and dependent on the organ in which they are expressed. Their detailed study and the affected signaling pathways may provide evidence for clinical treatment strategies. Below, we discuss some of the newer mechanisms of action by which circRNAs can induce resistance to conventional therapies in tumors. 

Many recent studies about circRNAs and chemoresistance involved colorectal cancer (CRC). About 40% of CRC shows mutations in KRAS gene, whose protein product plays an important role in cell proliferation signaling and chemoresistance. Diverse therapies in the field of precision oncology research targeting KRAS mutations have therefore been implemented [97]. To investigate the circRNA expression in isogenic CRC cell lines expressing wild-type and G13D mutant KRAS, Dou and colleagues performed RNA-Seq analysis [84]. Notably, they reported that mutant KRAS was responsible for the downregulation of circRNA expression at a global level, showing a widespread effect and that the regulation of circRNAs occurred independently of their host genes. Some of the identified circRNAs also participated in exosomal trafficking and became more abundant in exosomes than cells [84].

Microarray analysis of exosomal RNAs from chemoresistant and chemosensitive CRC cells showed a group of circRNAs with altered expression [54]. They were validated in CRC patients, and hsa_circ_0032883 was found to be significantly higher in serum exosomes of chemosensitive patients compared with chemoresistant patients [54]. This data suggested that hsa_circ_0032883 could be a tumor suppressor and a promising target for the response to chemotherapy. 

Very recently, several research groups have uncovered altered expressions of circRNAs in CRC, which, by supporting chemoresistance, induce cell proliferation, invasion, and migration through different mechanisms, the most common being that of miRNA sponging. For example, circ_0020095 was found to be highly present in CRC tissues and cells affecting proliferation, migration, invasion, and cisplatin resistance [55]. Mechanistically, circ_0020095 bound to miR-487a-3p, increasing SOX9 expression [55]. Xu and colleagues showed the interaction between circ-FBXW7 and miR-18b-5p in CRC cell lines [56]. circFBXW7 was found at low levels in oxaliplatin-resistant CRC patients and cells. Notably, circFBXW7 was secreted by circFBXW7-transfected cells by exosome transport and could be transferred to resistant CRC cells, leading to resistant cells becoming sensitive to oxaliplatin, increasing oxaliplatin-induced apoptosis, inhibiting oxaliplatin-induced EMT, and suppressing oxaliplatin efflux [56]. From the therapeutic point of view, circFBXW7 delivery by exosomes could inhibit chemoresistance to oxaliplatin in CRC by directly binding to miR-128-3p, speculating a precise therapeutic strategy for oxaliplatin-resistant CRC patients. 

An interesting, recent paper reported the involvement of circ_0005963 in the crosstalk between glycolysis and chemoresistance in CRC [57]. Tumors, including colorectal cancer, rapidly grow and hinder chemotherapy, usually trying to produce ATP through aerobic glycolysis where the M2 isoform of pyruvate kinase (PKM2) plays a pivotal role. In this research, the circular RNA hsa_circ_0005963 (ciRS-122) was found to act as sponge for PKM2-targeting miR-122 to induce chemoresistance. In vitro and in vivo results showed that exosomes from oxaliplatin-resistant cells delivered ciRS-122 to sensitive cells, and using the sponging mechanism previously cited, glycolysis and drug resistance was increased. Interestingly, exosomal ciRS-122 could inhibit glycolysis and reverse resistance to oxaliplatin by regulating the ciRS-122-miR-122-PKM2 pathway in vivo [57]. 

Very recently, the abnormal expression of circular RNAs related to tumor chemoresistance has exploded, showing multiple signaling pathways involved in tumor drug resistance. To give some examples, in breast cancer (BC) patients, hsa_circ_0025202 was found to inhibit cell tumorigenesis and Tamoxifen resistance via the miR-197-3p/HIPK3 axis, suggesting a potential therapeutic strategy to bypass chemoresistance in BC patients [75]. High expression of circGFRA1 was found to induce the resistance of triple negative breast cancer (TNBC) to Paclitaxel (PTX) by promoting the expression of miR-361-5p, which reduced TLR4 expression [58]. In Ovarian Cancer (OC), circNRIP1 downregulation could inhibit PTX resistance affecting the miR-211-5p/HOXC8 axis [59], while serum exosomal circular forkhead box protein P1 (circFoxp1) conferred cisplatin resistance by sequestering miR-22 and miR-150-3p, causing downregulation of CEBPG and FMNL3 and poor survival outcome [60].

Non-small cell lung cancer (NSCLC) patients have a low survival rate, and the advanced NSCLC cases that cannot receive molecular targeted therapy or immunotherapy are treated with the standard first-line regimen, where the combination treatment of carboplatin (CBDCA) plus paclitaxel (PTX) is one of the most widely used regimens, in which both agents are administered once every three weeks [98]. Several papers have recently reported the involvement of circRNAs in lung cancer, which are associated with tumor tissues, poor prognosis of patients and resistance to chemotherapeutic and immunotherapeutic agents.

Recently, high expression of hsa_circ_0002874 has been evidenced in NSCLC [61] and predicted an advanced TNM stage. Interestingly, hsa_circ_0002874 acts by sequestering miR-1273f, which targets the MDM2/p53 pathway [61]. Furthermore, the authors assessed the important role of hsa_circ_0002874/miR-1273f axis on PTX resistance in NSCLC. They constructed a drug-resistant xenograft mouse model by subcutaneous injection of A549 lung/Taxol-resistant cells, and mice were treated for 10 days with agomir-1273f plus PTX, siRNAs directed to circRNA (siRNAs-ciR) plus PTX, or PTX alone. The tumor size in the agomir-1273f group and siRNAs-ciR group were significantly smaller than those in the PTX group, highlighting the relevance of the hsa_circ_0002874/miR1273f/MDM2/p53 pathway on PTX resistance in vivo [61].

The expression of hsa_circ_0014235 was also reported to be significantly elevated in NSCLC serum-derived exosomes, tumor tissues and cell lines, and it was associated with tumor growth and resistance to cisplatin (CDDP) in vivo [99]. Hsa_circ_0001946 has been shown to induce the activity of the Nucleotide Excision Repair (NER) pathway in lung cancer cells by influencing resistance to CDDP [62]. XPA, XPC, Rad23B, RPA14, RPA32, RPA70, and ERCC1, which are NER pathway-related genes, were induced after the knockdown of hsa_circ_0001946 expression. The downregulation of hsa_circ_0001946 expression activated the NER signaling pathway, which, in parallel, decreased sensitivity to CDDP and increased DNA repair [62].

Hepatocellular carcinoma (HCC) is the most common disease of primary liver cancer (about 80%) and is the fourth leading cause of cancer-related death worldwide [100]. Sorafenib is a multi-kinase inhibitor that exhibited quite a good clinical efficacy in HCC patients; however, poor prognosis of HCC is closely related to the development of acquired resistance [101]. Using RNA-seq analysis performed in sorafenib-resistant HCC cell lines (SR-HepG2 and SR-Huh7), Weng and colleagues compared the circRNA profiles of sorafenib-resistant and sorafenib-sensitive cells, identifying the 269 most differentially expressed circRNAs [102]. Hsa_circ_0025039 (circFOXM1) was found to be closely associated with the best score for sorafenib-resistant cells. Notably, circFOXM1 expression was higher in the tumor and adjacent tissues than in adjacent non-tumoral tissues, and its expression was higher in sorafenib-resistant HCC tissues than in sorafenib-sensitive tissues [102].

Natural killer (NK) cells play a pivotal role in the innate antitumor immune response. Diverse groups assessed that NK cell dysfunction is present in various malignant tumors, including HCC [103].

An interesting paper recently reported high circUHRF1 associated with human HCC tissues [63]. These aberrant circUHRF1 levels are related with poor prognosis and NK cell dysfunction in HCC patients. The high level of circUHRF1 was also documented in exosomes purified from the plasma of HCC patients, and it was associated with decreased NK cell proportion and decreased NK cell tumor infiltration [63]. Furthermore, the authors showed that circUHRF1 may drive resistance to anti-PD1 immunotherapy in HCC patients, establishing a potential therapeutic strategy for HCC patients. Huang and colleagues also showed a crosstalk between circMET (hsa_circ_0082002), which was overexpressed in HCC tissue samples, and cancer immuno tolerance [46]. Mechanistically, circMET induced HCC development and immune tolerance via the Snail/DPP4/CXCL10 axis using in vitro and in vivo experiments, determining resistance to anti-PD1 treatment [46].

The circRNA functions in the drug response reviewed in this paragraph are summarized in Table 1.

## 7. Conclusions

Due to the higher stability of circRNAs with respect to miRNAs and lncRNAs, shown by the covalently closed ring structure, circRNAs cannot be degraded by most ribonucleases compared with linear RNAs [104]. Furthermore, deregulated circRNA expression is significantly associated with cancer and could have clinical significance as a diagnostic and prognostic biomarker, as well as an evaluator of the therapeutic efficacy of cancer treatments. Some intron-containing circRNAs remain in the nucleus and can act as regulators of parental genes, whereas others are transferred to the cytoplasm and may play a role as miRNA sponges and protein sponges or can be translated into proteins or peptides [105]. However, there are still a discrete number of unknown circRNAs with unexplored functions that could have interesting clinical applications. Interestingly, the majority of circRNAs are also incorporated into exosomes, strongly enriching their presence in plasma compared with cancer tissues to perform functions away from the primary tumor. 

In many cancer patients, early symptoms are not easily identifiable, as happens in ovarian carcinoma or gastric cancer, and without robust and specific biomarkers for early diagnosis, patients experience disease progression and further development of metastases. To date, a variety of ncRNAs, as miRNAs and circRNAs, are emerging as powerful biomarkers for early diagnosis and prognosis prediction, especially due to their presence and easy detection in body fluids [106,107,108]. The extensive study of these ncRNAs in large cohorts of patients and in various biological samples will hopefully strongly contribute to the development of novel diagnostic/prognostic strategies that will help cancer management.

## Figures and Tables

**Figure 1 cancers-13-03154-f001:**
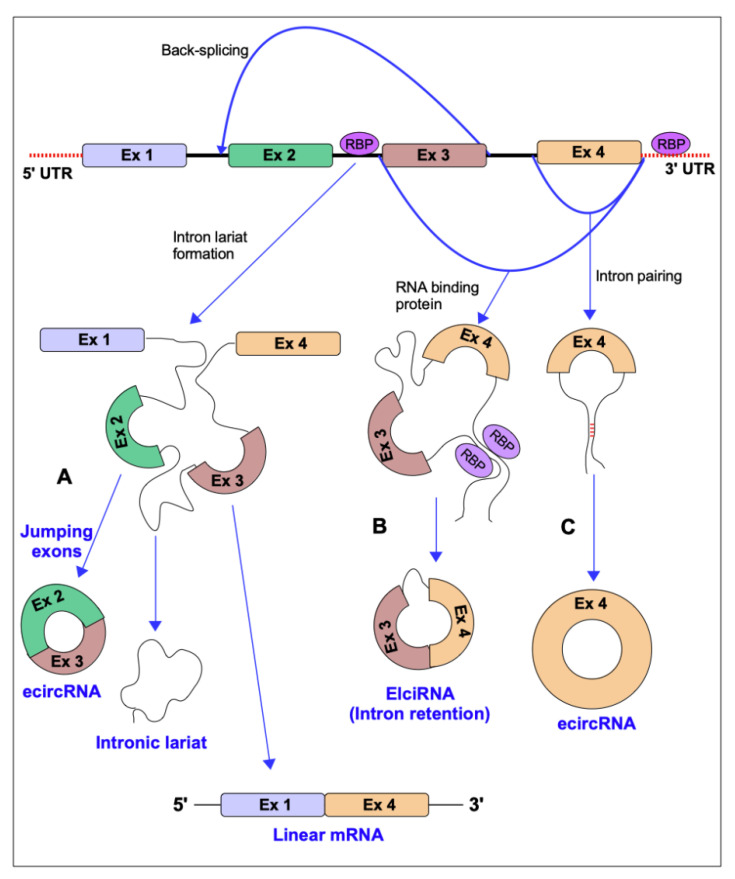
The circular RNA (circRNAs) biogenesis. Three mechanisms that lead to circRNA biogenesis are briefly described in the picture. (**A**) The first mechanism causes exon skipping and intron lariat production. “Jumping exons” consist of bringing together two distant exons with the formation of intron circular lariat. The lariat formation allows for the circularization of the “skipped” exons. Three different types of RNA molecules are produced: circRNA, intron lariat, and linear mRNA containing skipping exons. (**B**) The second mechanism is regulated by RNA-binding proteins, which bind the neighboring introns of the exon that should circularize and create an RNA loop when RBPs dimerize. The link between 3′ and 5′ ends of the circularized exons can facilitate splicing. (**C**) In the third mechanism, the circularization of the exon/exons is promoted by the complementary pairing of the flanking introns. Pairing with a complementary inverted sequence, such as ALU repeats, could increase back-splicing.

**Figure 2 cancers-13-03154-f002:**
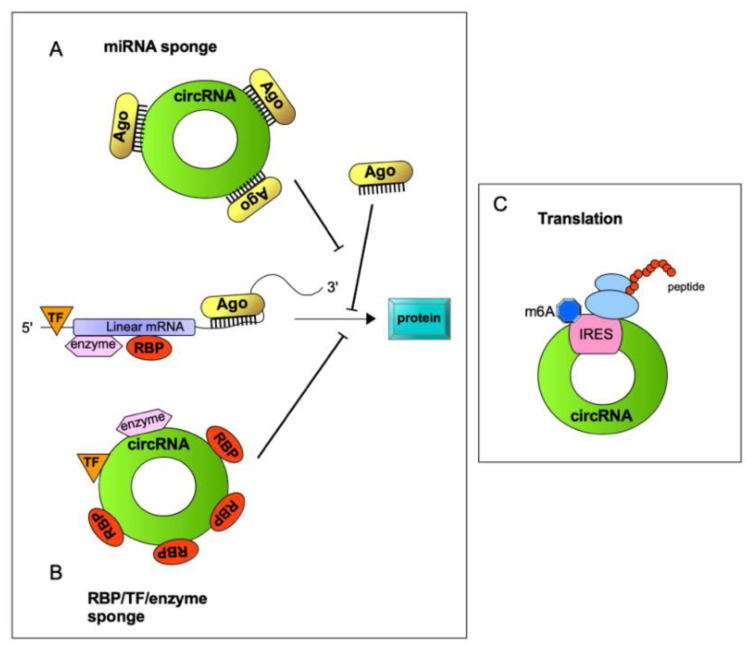
The biological functions of circRNAs. (**A**) circRNAs can function as a miRNA sponge to rescue target mRNA expression, thus inducing translation; (**B**) some circRNAs bind to RNA binding proteins (RBP), transcription factors (TF) or enzymes altering gene expression and cellular metabolism; (**C**) some circRNAs contain an open reading frame (ORF) and can be translated, producing peptides. UTRs including m6A modification, IRES, and/or IRES-like sequences may induce circRNA translation.

**Figure 3 cancers-13-03154-f003:**
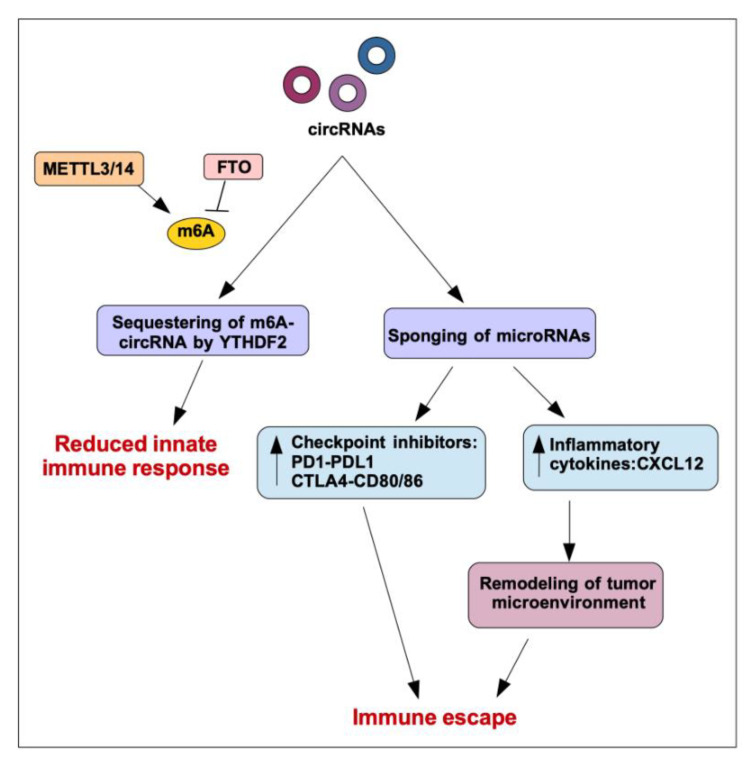
Impact of circRNAs on translatiuon and immune functions. The N6-methyladenosine (m6A) RNA modification of human circRNA, promoted by the activity of METTL3/14 while counteracted by FTO, leads to the inhibition of the innate immune response, which is based on the subtraction of modified circRNAs by the reader m6A YTHDF2. In the context of cancer, circRNAs impact immune function due to their miRNA sponging activity. In fact, the inhibition of microRNAs leads on the one hand to the induction of checkpoint inhibitors and, on the other, to the released expression of soluble inflammatory mediators, such as CXCL12. Both of these activities promote the depletion of T lymphocytes and immune escape.

**Table 1 cancers-13-03154-t001:** Summary of oncogenic and tumor suppressors reviewed circRNAs.

circRNAs	Targets or Biological Output	Molecular Function	Cancer Type	Ref.
circAmotl1	c-Myc	Interaction with proteins and to function as protein decoys	Breast	[18,19]
	Stat3			
circFoxo3	ID-1	Interaction with proteins and to function as protein decoys		[20]
	FAK			
	HIF-1a			
	E2F1			
circPABPN1	PABPN1	Interaction with proteins and to function as protein decoys	Cervical	[21]
circPCNX	AUF1	Interaction with proteins and to function as protein decoys	Cervical	[22]
circMALAT1	PAX5	Enhancing protein complexes formation and/or activity	Hepatocellular	[23]
circNSUN2	IGF2BP2	Enhancing protein complexes formation and/or activity	Colorectal	[24]
	HMGA2			
circYAP	YAP	Enhancing protein complexes formation and/or activity	Breast	[25]
	eIF4G			
	PABP			
circPOK	ILF2/3	Enhancing protein complexes formation and/or activity	Sarcoma	[26]
	IL-6			
	VEGF			
circFoxo3	MDM2	Interaction between different proteins	Breast	[20]
	p53			[27]
	p21			[27]
circCcnb1	H2AX	Interaction between different proteins	Breast	[28]
	Wild type-p53			
	Bclaf1			
circANRIL	PES1	Interaction between different proteins		[29]
circGSK3β	β-catenin	Interaction between different proteins	Esophageal	[30]
circDONSON	SOX4	Transcriptional regulation	Gastric	[31]
			Liver	
circHOT1	TIP60	Transcriptional regulation	Hepatocellular	[32]
	NR2F6			
circEIF3J	U1 snRNP	Control of the expression of parental genes: control of transcription initiation	Cervical	[33]
circPAIP2	U1 snRNP	Control of the expression of parental genes: control of transcription initiation	Cervical	[33]
circXIAP	FUS	Control of the expression of parental genes: control of transcription initiation	Prostate	[34]
	XIAP			
FECR1	TET1	Control of the expression of parental genes: control of transcription initiation	Breast	[35]
	FLI1			
circCUX1	EWSR1	Control of the expression of parental genes: control of transcription initiation	Neuroblastoma	[36]
ci-ankrd52	ankrd52	Control of the expression of parental genes: control of transcription elongation	Cervical	[37]
Hsa_circ_0075804	HNRNPK	Regulation of protein scaffold to impact on the translation	Retinoblastoma	[38]
circBACH1	HuR	Regulation of protein scaffold to impact on the translation	Hepatocellular	[39]
circZNF609	translated	Translation		[40]
circβ-catenin	β-catenin-370aa	Translation	Hepatocellular	[41]
circFBXW7	FBXW7-185aa	Translation	Glioma Breast	[42,43]
circ_0020397	miR-138	Regulation of immune functions	Colorectal	[44]
circ_0020710	miR-370-3p	Regulation of immune functions	Melanoma	[45]
circMET	miR-30-5p	Regulation of immune functions	Hepatocellular	[46]
CDR1-AS	miR-7	Functions in exosomes	Brain	[47,48]
circPDE8A	EMT	Functions in exosomes	PDAC	[49]
circIARS	RhoA	Functions in exosomes	Pancreatic	[50]
	RhoA-GTP			
circ_17720	cancer recurrence	Functions in exosomes	Prostate	[51]
circ_178252				
circ_115617				
circ_14736				
circ_30029				
circ_117300				
circ_176436				
circ_112897				
circSATB2	miR-326	Functions in exosomes	Lung	[52]
circWHSC1	miR-145	Functions in exosomes	Ovarian	[53]
Hsa_circ_0032883	chemosensitivity	Drug resistance	Colorectal	[54]
Hsa_circ_0020095	miR-487a-3p	Drug resistance	Colorectal	[55]
circ-FBXW7	miR-18b-5p	Drug resistance	Colorectal	[56]
circ_0005963	miR-122	Drug resistance	Colorectal	[57]
circGFRA1	miR-361-5p	Drug resistance	Breast	[58]
circNRIP1	miR-211-5p	Drug resistance	Ovarian	[59]
circFoxp1	miR-22	Drug resistance	Ovarian	[60]
	miR-150-3p			
Hsa_circ_0002874	miR-1273f	Drug resistance	Lung	[61]
Hsa_circ_0001946	NER signaling pathway	Drug resistance	Lung	[62]
circUHRF1	Resistance to anti-PD1 immunotherapy	Drug resistance	Hepatocellular	[63]

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
