# Peer review of "New Molecular Mechanisms and Clinical Impact of circRNAs in Human Cancer"

_cancers, 2021, doi:10.3390/cancers13133154_

Round 1
Reviewer 1 Report
The authors wrote a timely review on an interesting topic. In particular, This review aims to discuss emerging circRNA functions, beyond miRNA sponging activity, such as circRNAs binding to specific RNA binding proteins (RBPs), sequestering specific protein factors, encoding proteins/peptides that are involved in carcinogenesis and metastatization.
They should make some revisions to the text:
1) Please rewrite in 2-3 separate statement the following: Interestingly, through new RNA sequencing methodologies and new bioinformatics approaches, circRNAs have forcefully emerged in the clinical and basic research landscape [1, 5, 6], furthermore highlighting that human transcriptome includes more than 180,000 circRNAs [7] and that their expression pattern varies among cell types, diseases and developmental stages of the living beings, including plants and invertebrates [8-11].
2) Table 1: the authors reported the targets, but not the function: please clarify
3) the authors should delete the description of the epidemiology of different tumors, such as "Non-small cell lung cancer account 80% of all lung cancer...". This review is going to be published to a journal in the Cancer field and such statements are useless.
4) Please revise: The patients with advanced NSCLC that don’t receive molecular targeted therapy or immunotherapy, may be treated with cytotoxic chemotherapy by Paclitaxel (PTX) [95].": first do not use "don't", second clarify which regimen include paclitaxel.
5) the figures are nice, but I am missing a table summarizing the most important findings in the "Functions of circRNAs in the Exosomes".
6) some of the paragraphs are too long and the authors are losing the focus on the most important message they would like to convey: please try to make always a clear summary of the most important results at the end of each paragraph
7) the authors should consider adding (when describing pancreatic cancer) a recent review on circular RNA published on Cancers (Basel). 2020 Nov 4;12(11):3250. doi: 10.3390/cancers12113250.
PMID: 33158116
Author Response
We wish to thank the reviewer for the revision which improved the manuscript. We have appreciated the comments and we revised the manuscript accordingly.
Below, please find the detailed list of our responses and the changes we made in the revised version of the manuscript marked in red color.
The authors wrote a timely review on an interesting topic. In particular, This review aims to discuss emerging circRNA functions, beyond miRNA sponging activity, such as circRNAs binding to specific RNA binding proteins (RBPs), sequestering specific protein factors, encoding proteins/peptides that are involved in carcinogenesis and metastatization.
They should make some revisions to the text:
- Please rewrite in 2-3 separate statement the following: Interestingly, through new RNA sequencing methodologies and new bioinformatics approaches, circRNAs have forcefully emerged in the clinical and basic research landscape [1, 5, 6], furthermore highlighting that human transcriptome includes more than 180,000 circRNAs [7] and that their expression pattern varies among cell types, diseases and developmental stages of the living beings, including plants and invertebrates [8-11].
We have rewritten this sentence as suggested.
- Table 1: the authors reported the targets, but not the function: please clarify
We thank the reviewer for the comment. We added two columns to Table 1 describing the functions of each circRNA and the type of tumor in which they are described in the reference.By this way we are sure that the revised Table 1 will be more informative.
3) the authors should delete the description of the epidemiology of different tumors, such as "Non-small cell lung cancer account 80% of all lung cancer...". This review is going to be published to a journal in the Cancer field and such statements are useless.
4) Please revise: The patients with advanced NSCLC that don’t receive molecular targeted therapy or immunotherapy, may be treated with cytotoxic chemotherapy by Paclitaxel (PTX) [95].": first do not use "don't", second clarify which regimen include paclitaxel.
We reformulated the concepts of these sentences according to the reviewer suggestions in the points 3 and 4.
5) the figures are nice, but I am missing a table summarizing the most important findings in the "Functions of circRNAs in the Exosomes".
The reviewer's observation was correct, and in the new version of the manuscript we have described the circRNA functions in the exosomes in the new column of the Table 1.
6) some of the paragraphs are too long and the authors are losing the focus on the most important message they would like to convey: please try to make always a clear summary of the most important results at the end of each paragraph.
We thank the reviewer for this suggestion.We have made changes to the text to follow his /her directions.
7) the authors should consider adding (when describing pancreatic cancer) a recent review on circular RNA published on Cancers (Basel). 2020 Nov 4;12(11):3250. doi: 10.3390/cancers12113250.
PMID: 33158116
I thank the reviewer for this interesting report that we have included in the text (new ref. 72).
Reviewer 2 Report
In this manuscript, Fontemaggi and colleagues discuss the emerging roles of circular RNAs in human cancers. The manuscript is well organized and systematic. However, I have a few comments to improve the manuscript.
- The table may include further information such as cancer type, circRNA target (miRNA/RBP) and the downstream target gene, and function. It might be better to have subsections in the table by dividing the list into miRNA sponge, RBP sponge, circRNA translation, circRNA in exosomes, etc.
- Figure 1; Exon 1 is the starting of any RNA. There should not be any intron (line) upstream to Exon1 at the 5’ end. Please edit the figure accordingly.
- The figures for circRNA biogenesis and function are showing the general mechanisms which are extensively reviewed in other manuscripts. The authors may include figures showing the function of circRNAs in human cancers.
Author Response
We wish to thank the reviewer for giving us the opportunity to revise and improve the manuscript appreciating the comments. We revised the manuscript accordingly.
Below, please find the detailed list of our responses and the changes we made in the revised version of the manuscript marked in red color.
In this manuscript, Fontemaggi and colleagues discuss the emerging roles of circular RNAs in human cancers. The manuscript is well organized and systematic. However, I have a few comments to improve the manuscript.
- The table may include further information such as cancer type, circRNA target (miRNA/RBP) and the downstream target gene, and function. It might be better to have subsections in the table by dividing the list into miRNA sponge, RBP sponge, circRNA translation, circRNA in exosomes, etc.
We thank the reviewer for this suggestion which will surely increase the understanding of the manuscript.We have integrated Table 1 into the new version of the text.
- Figure 1; Exon 1 is the starting of any RNA. There should not be any intron (line) upstream to Exon1 at the 5’ end. Please edit the figure accordingly.
We thank the reviewer for this observation.We accordingly modified Figure 1 by changing the design for the 5 'and 3' UTR sequences.
- The figures for circRNA biogenesis and function are showing the general mechanisms which are extensively reviewed in other manuscripts. The authors may include figures showing the function of circRNAs in human cancers.
To address this specific point, we have added new Figure 3 which describes some important functions of circRNAs in the mechanisms of immune response modulation in cancer.
Reviewer 3 Report
This review article by Fontemaggi et al. gives a nice update of recent findings about molecular mechanisms of circRNAs regulation and their function in cancer pathology. While focusing mainly on novel mechanisms of circRNAs’ actions, which are different than miRNA sponging activity, the authors discuss the involvement of circRNAs in cancer immunology, exosomes and drug resistance. In light of an increasing number of reports suggesting potential of circRNAs as cancer biomarkers and their possible involvement in cancer therapeutic intervention, this review will definitely help the reader to understand the importance of circRNAs in cell function, their impact on cancer cell abnormality, and possibility for clinical application.
The authors did a great work describing the function of each circRNA in the text, however they need to revise Table 1 to summarize their discussion in a more systematic way.
In particular, in Table 1, for each cited circRNA, they should make separate columns specifying CircRNA, Target Molecule, Function, Cancer Type and References.
The other specific points that need to be addressed are:
- References 37 and 81 should be included in Table 1
- Reference 92 is not cited for CircNRIP1 in Table 1
- Reference 93 is twice in Table 1 (incorrectly for CircNRIP1)
Author Response
We wish to thank the reviewer for giving us the opportunity to revise and improve the manuscript appreciating the comments. We revised the manuscript accordingly.
Below, please find the detailed list of our responses and the changes we made in the revised version of the manuscript marked in red color.
This review article by Fontemaggi et al. gives a nice update of recent findings about molecular mechanisms of circRNAs regulation and their function in cancer pathology. While focusing mainly on novel mechanisms of circRNAs’ actions, which are different than miRNA sponging activity, the authors discuss the involvement of circRNAs in cancer immunology, exosomes and drug resistance. In light of an increasing number of reports suggesting potential of circRNAs as cancer biomarkers and their possible involvement in cancer therapeutic intervention, this review will definitely help the reader to understand the importance of circRNAs in cell function, their impact on cancer cell abnormality, and possibility for clinical application.
The authors did a great work describing the function of each circRNA in the text, however they need to revise Table 1 to summarize their discussion in a more systematic way.
In particular, in Table 1, for each cited circRNA, they should make separate columns specifying CircRNA, Target Molecule, Function, Cancer Type and References.
We thank the reviewer for the comments. We added two columns to the Table 1 describing the functions of each circRNA and the type of tumor in which they were described in the reference. By this way we are sure that the revised Table 1 will be more informative.
The other specific points that need to be addressed are:
- References 37 and 81 should be included in Table 1
- Reference 92 is not cited for CircNRIP1 in Table 1
- Reference 93 is twice in Table 1 (incorrectly for CircNRIP1)
Accordingly to these specific points, we have integrated and corrected the Table 1.